# Tunable Device for Long Focusing in the Sub-THz Frequency Range Based on Fresnel Mirrors

**DOI:** 10.3390/mi15060715

**Published:** 2024-05-29

**Authors:** Giancarlo Margheri, Tommaso Del Rosso

**Affiliations:** 1Institute for Complex Systems of National Council of Researches of Italy, Via Madonna del Piano, 50019 Sesto Fiorentino, Italy; 2Department of Physics, Pontifícia Universidade Católica do Rio de Janeiro, Rua Marques de São Vicente, Rio de Janeiro 22451-900, Brazil; tommaso@puc-rio.br

**Keywords:** long-focus devices, beam shaping, THz devices

## Abstract

THz radiation has gained great importance due to its potential applications in a wide variety of fields. For this reason, continuous efforts have been made to develop technological tools for use in this versatile band of the electromagnetic spectrum. Here, we propose a reflecting device with long focusing performances in the sub-THz band, using a bimirror device in which the relative angle is mechanically adjusted with the displacement of one of the mirrors. Despite the simplicity of the setup, the performance of this device is satisfactory down to a frequency of 0.1 THz. Theory and experience confirm that the bimirror is capable of focusing 0.1 THz radiation with a 2× magnification of the maximum input intensity while maintaining a longitudinal full width at half maximum (FWHM) of about 6 mm, which is about 12 times the depth of focus of a cylindrical optical element of the same focal length. In the absence of suitable THz equipment, the invariance property of the Fresnel diffraction integral allowed the predicted behavior to be tested in the THz range using conventional equipment operating at visible frequencies.

## 1. Introduction

Recently, the enormous benefits of using electromagnetic radiation at terahertz frequencies (100 GHz to 10 THz) have been recognized. Indeed, a wide range of potential applications have been proposed in biomedicine [1], non-destructive industrial control [2], wireless communications [3], sensing [4], food quality [5], and postal security screening [6]. Because of this versatility, the handling of radiation in this frequency range is a key issue in the implementation of its practical use. However, low-cost and high-performance optical tools still suffer from quality and cost problems [7,8]. Among various optical functions, long-focusing devices have attracted much attention due to their particular ability to maintain a quasi-constant cross-sectional distribution profile due to their robustness to diffraction, which becomes a non-trivial problem at wavelengths in the THz range. In this sense, maintaining a narrow focus over long distances becomes a significant challenge. Several solutions have been studied and experimentally demonstrated using various techniques (computer numerical control manufacturing, injection molding, ultrashort pulse laser ablation, milling, and photomicrolithography), but they still suffer from problems of cost and complexity or work only in a limited frequency range. A particularly attractive tool that is rapidly overtaking other techniques and represents the best compromise between cost and benefit is 3D printing [9], which has received much attention and is a rapidly growing technology. On the other hand, its current resolution cannot produce details smaller than 0.1 mm, making these elements prone to scattering that can only be avoided by working at wavelengths higher than 0.5 mm (a 0.6 THz frequency). Similar benefits have been obtained by resorting to conventional cost-effective molding techniques that permit the efficient focusing of THz radiation [10]. Among various issues, bulk material transmission is also a feature that can limit the performance of THz radiation [11]. 

Almost all the proposed solutions are well suited to work in a restricted range of frequencies as they are not tunable or their tunability is extended to no more than 0.5 THz [12]. In this sense, promising results have been obtained in recent years using metallic devices. Plate waveguides have been proposed to perform several functions by exploiting the good transmission properties of radiation in the THz regime and the phase change introduced by shaped profiles. These devices ensured indeed long focusing, but the reported operational frequency range is limited to 0.1 THz [13]. The claimed tunability is indeed possible, but a proper redesign and fabrication of the basic components seem to be unavoidable even if the design tolerances, as pointed out therein, do not require major efforts thanks to the wavelengths implied (ranging from some mm to hundreds of micrometers). 

Another solution based on the free propagation of the electromagnetic field, implying the use of metallic components, was recently proposed by us [14]. The device uses a metal-coated PDMS layer whose huge optothermal deformation, which is produced by a focused pump laser, allows it to manage optical path differences that fall into the THz range. In that work, we demonstrated a device based on a reflecting cylindrical bimirror that is able to produce virtual foci operating at frequencies between 1 and 10 THz. However, the operation at lower frequencies is hampered by the limit of expansion of the polymer, which suffers degradation at the high temperatures needed to reach usable deformations. Indeed, even if the reflecting metal layer undergoes stable swelling that would permit a sub-THz application, in practice, this feature cannot be practically exploited because of the scarce reproducibility of the thermal threshold, which does not guarantee the fabrication of reliable devices. On the other hand, working in this THz window is highly desirable as many of the available sources, especially those developed in compact versions, work in this frequency range; see, for instance, the interesting review in Ref. [15].

In order to eliminate this limitation, in this work, we explored the possibility of long-focusing sub-THz radiation by fabricating a device composed of two planar mirrors arranged to form an adjustable dihedron, which is a characteristic that allows for the accumulation of phase changes large enough to produce appreciable changes in the propagation of sub-THz radiation. 

However, it is worth noticing that the high wavelengths involved introduce strong diffraction effects that can a priori partially counterbalance the mechanism that produces the long-focusing effect, so an in-depth evaluation of the electromagnetic propagation is needed to establish the possible advantages and limitations of this approach.

Indeed, we found that this system can obtain a long focus, both real and virtual (whereas in [14], only a virtual focus is achievable) in the whole sub THz regime (0.1–1 THz). This device is well known in two-beam interferometry in the-Vis–NIR range, but to our knowledge, this work is an original proposal to manage sub-THz radiation to produce efficient focusing in spite of the high diffractive effects.

The bimirror can form foci with tunable features (depth of focus and width) and can be implemented in an easy way, making it possible to operate it flexibly in the whole sub-THz range. 

## 2. Theory

### 2.1. Working Principle

The operational description of the device is depicted in Figure 1a below.

Two mirrors of width a, separated by a gap s, are folded to form a wedge angle α with respect to the input plane. The rays incident on the mirrors are reflected at angles 2α and −2α with respect to the y-axis and thus interfere, forming a rhomboid superposition zone. However, an electromagnetic calculation is required to correctly characterize the device behavior, as ray tracing does not take into account either the interference or diffraction effects.

Referring to Figure 1b, where the geometric characteristics are shown, the input field is a Gaussian electric field with unit amplitude propagating in the y-direction, given by E(x_i_) = exp(−(x_i_/w_0_)^2^) exp(iky), where x_i_ is the input abscissa in the input plane, w_0_ = 5 mm is the beam waist (which, unless otherwise specified, remains constant throughout the paper) k = 2π/λ is the wavevector modulus, and λ is the wavelength in vacuum. 

The output abscissa, x_o_, is generally different from x_i_, but as long as α is small enough, the relative difference |x_o_ − x_i_|/x_i_, which is equal to tg(2α), is small, and the two values can be assumed to be approximately equal. In this work, this difference reaches 11% in the worst case (α = 0.27 rads, see Experimental section) and we will therefore consider this thin device approximation as valid. 

Moreover, the optical path difference of the input wave with respect to the wave that crosses the device and is reflected by it is approximately equal to twice the optical path from the input plane to the device surface, namely OPD(x) ≈ 2(a − s/2 − |x_i_|)·α, giving a total phase variation Φ(x) = 2π/λ·OPD(x). The output field can be calculated using the Fresnel integral in cylindrical symmetry [16]: (1)$$U(x,y)=\frac{A}{\sqrt{\lambda y}}\int E(x_{i})\cdot \exp(-i\cdot \frac{4\pi }{\lambda }\left|x_{i}\right|\cdot \alpha )\cdot \exp(i\cdot \frac{\pi \cdot x_{i}}{\lambda y})dx_{i}$$
and the intensity is given by:(2)$$I(x,y)=\frac{{\left|U(x,y)\right|}^{2}}{\lambda y}$$

In expression (1), we have made explicit the total OPD(x) in the first phase term and set A = exp(i·(2a − s)) outside the integral as it is x_i_-independent. As noted in [14], the integral is invariant in the transverse x coordinate if the following apply: (a) the ratio α/λ and (b) the product λ·y are held constant; and (c) the input field has an established transverse spatial distribution. This observation has allowed us to experimentally verify the focusing property of the bimirror using visible radiation. For example, the field intensity at the THz wavelength λ, can be obtained with a wedge angle α and a y-coordinate calculated with the relationships α/λ = α_exp_/λ_exp_ and λ·y = λ_exp_·y_exp_, where λ_exp_, y_exp_ are the experimental wedge and focal position obtained at a given test wavelength λ_exp_. The corresponding values of α and y at wavelength λ are therefore α = α_exp·_ M_λ_ and y = y_exp_/M_λ_, where M_λ_ = λ/λ_exp_ is now referred to as the spectral amplification factor. With these substitutions, it is easy to verify that the intensity measured at coordinates (x, y_vis_) at wavelength λ_vis_, using a Gaussian field input with waist w_0_ is equal to that obtained using radiation with the same input field and wavelength λ at coordinates (x, y).

The theoretical modeling was performed using finite element modeling software, namely the commercial software COMSOL 5.3a. 

It is worth noting that in any electromagnetic simulation using Finite Element Modeling (FEM), the wavelength must always be resolved by the mesh in order to find an accurate solution to Maxwell’s equations. This requirement makes it difficult to simulate models that are large relative to the wavelength. There are several methods for stationary wave optics problems that can handle large models. These include the so-called diffraction formulas, such as the Fraunhofer, Fresnel–Kirchhoff and Rayleigh–Sommerfeld diffraction formulas, and the beam propagation methods (BPMs), such as the paraxial BPM and the angular spectrum method. 

In this case, the output field was calculated using the Electromagnetic Wave Beam Envelope module of the COMSOL 5.3a software in a 2D frame, which is infinite and invariant in the z-direction. This is a special modeling capability designed specifically for solving large domain electromagnetic problems using FEM. The wave equation is first transformed into a Helmholtz equation by considering a monochromatic wave. The electromagnetic field is then solved by searching for solutions in the form E_s_ = E(x,y)·exp(i·Φ(x,y)), where the phase Φ(x,y) is chosen to be close to that of a plane wave traveling in the y-direction, namely Φ(x,y) ≈ (2π/λ)·y, and E(x,y), called the envelope function, is usually a slowly varying function. With these assumptions, it is possible to solve the problem with high accuracy and reasonable computational times by choosing a not too dense mesh. This makes it possible to simulate large wave optics problems even with limited computing power. In the model shown in Figure 1b, the input boundary must be matched to the input field with the correct amplitude and phase distributions, represented by the field E_i_ in Figure 1a, by imposing the Matched Boundary Condition (MBC) on the Helmholtz equation. An appropriate mesh is then selected until a suitable convergence of the solution is obtained. The light intensity, namely the modulus of the Poynting vector, is then obtained as one of the quantities calculated by the software. 

The rectangular model space was surrounded by three absorbing phase-matched layers (PMLs), which absorb the outgoing radiation. Scattering boundary conditions were imposed on their outer boundaries to enhance the extinction of the outgoing radiation quenched by the PMLs. However, in order to verify the correct predictions of the FEM model, the results were compared with those obtained using a routine implemented in the MATHCAD 2000 software that calculates the Fresnel diffraction integral of Equation (1). Excellent agreement was found for all the models tested.

### 2.2. Simulation Results

First, we have investigated the focusing properties versus the wedge angle *α =* 0, 0.1, 0.15, 0.2, rads at the two wavelengths λ *=* 3 mm (0.1 THz) and λ *=* 1.5 mm (0.2 THz). The results of these cases are shown in the plots of Figure 2. 

These plots show that there is an axial coordinate at which the intensity reaches an absolute maximum, which is referred to as the bimirror equivalent focal length f_eq_. At this position, the intensity in the transverse plane has a full width at half maximum (FWHM), which will be referred to as the focal width w. 

As a general trend, the decrease in α corresponds to an increase in the FWHM of the axial intensity, referred to as the depth of focus (DOF), which is in agreement with predictions based on geometric optics [17]. At the same time, there is a decrease in the focus width w and an increase in the maximum intensity, but this increase decreases at higher values of f_eq_. In all cases, the peak intensity on the axis is increased with respect to that of the input beam, and we characterize this increase by the focusing factor η = I_max_/I_0_, where I_max_ is the maximum axial intensity and I_0_ is the intensity on the axis of the input Gaussian beam. 

For example, consider the radiation at 0.1 THz, η = 2.3 for α = 0.2 rads and η = 1.6 for α = 0.1 rads. Like the DOF, the f_eq_ increases as α decreases. For example, considering α = 0.2 rads and α = 0.1 rads, the focal length f_eq_ goes from 4.8 to 6.9 mm, while the focus extension DOF goes from 12.2 to 24.6 mm. This can be a problem for effective use in real applications, as the instrument would then be placed too close to the samples to be analyzed. However, this drawback can be overcome by using a suitable relay optic, whose optical quality is not a critical issue due to the long wavelengths involved. 

We then studied focusing at fixed angles of incidence and varying the wavelength of the radiation, the results of which are shown in Figure 3.

At small angles, there are significant differences in the focal positions. For example, in the example in Figure 3a, the focal f_eq_ at λ = 1 mm (0.33 THz) is 5.4 mm, which is 2.3 times higher than the 4.8 mm calculated at 0.1 THz. As can be seen, this valuable difference is significantly reduced at higher angles and becomes practically negligible at higher frequencies. The figures show the DOFs and compare them with those, DOF_GO_, that can be calculated in the geometric optics approximation with the relation DOF_GO_ = w_0_/tg(2α). While at small angles DOF_GO_ is similar to the value of DOF at lower frequencies, this behavior is reversed at higher angles α. These variations can be as large as 57% (α = 0.2 rads and λ = 2 mm), so a geometrical description in the entire sub-THz frequency range is a good approximation of the device behavior when wedge angles are set in the range ~0.1 ÷ 0.15 rads. Although more detailed investigations are required to explain these peculiar differences, it seems likely that these deviations are due to diffraction at low frequencies and interference at high frequencies. 

As explained earlier, the wedge angle α is the key feature that determines an effective and long focusing effect. In a typical design path, the focusing factor η is fixed while trying to maximize the DOF at the same time. This desire has a limitation anyway, because for a given frequency and input waist, a focusing factor η uniquely defines a pair of values (α_η_, DOF_η_). This finding is repeated in the plots of Figure 4, which represent the situation where η = 2.

For example, a Gaussian input field frequency of 0.4 THz is focused with η = 2 when a wedge angle is set to α_2_ = 0.028 rads. The corresponding depth of focus is DOF_2_ = 83 mm. 

It is possible to increase η by reducing the wedge angle, but the DOF will be reduced at the same time. For example, calculations performed for η = 3 show that in the whole sub-THz range, the wedge α is in the range 0.028 ÷ 0.24 rads, while the DOF correspondingly goes from 80 to 9 mm. 

The wedge angle decreases non-linearly at higher frequencies, while the DOF_η_ increases as expected, and at higher frequencies, α_η_ decreases very slowly and becomes approximately independent of frequency from 0.6 to 1 THz. 

The long focusing property of the bimirror with equivalent focal length f_eq_ can be better appreciated by considering the performance of a cylindrical optical element (COE) with focal length f_eq_. In particular, we compare the depth of focus of the COE, calculated with the Gaussian optics formula as DOF_G_ = 2·π·w_G_^2^/λ, namely the double of the Rayleigh range, and the focus width w_G_ = 2·λ·f_eq_/(π·w_0_). 

The results of the calculations are shown in Table 1 and Table 2 for the cases α = 0.2 rads and 0.1 rads. It can be seen that the DOF of the bimirror is always higher than the DOF_G_, while the width w is wider than the w_G_ and decreases with increasing frequency. Note that at lower frequencies, w_G_~w.

For example, for the case of a frequency of 0.1 THz and f_eq_ = 4.8 mm, the calculations give a DOF_G_ = 1.9 mm (compared to a DOF = 12.2 mm) and w_G_ = 1.92 mm (compared to w = 1.8 mm). Thus, although the DOF of the COE is about 6.4 times smaller than that of the bimirror, the devices produce foci of similar width. Increasing the frequency to 0.66 THz gives DOF/DOF_G_ = 53.7 and w/w_G_ = 2. Comparing the two tables, we can also see that the focusing factor η increases with α and frequency. The first effect, which can also be seen from the plots in Figure 2, is due to the confinement of the radiation in a smaller volume, as clearly shown in the 2D plots in Figure 2, while the second effect is due to diffraction, which spreads the radiation more easily at lower frequencies, causing a loss of power confinement and a consequent reduction in the maximum intensity. This last observation provides quantitative evidence of the diffraction effects that can significantly reduce the focusing efficiency and underscores the need for careful analysis of wave propagation when very low frequencies are involved. We have also studied the effect of the finite gap between the mirrors on the focusing efficiency. In order to construct a tunable system, the mirrors must be moved with respect to each other, and any contact between them must be avoided in order to exclude the possibility that boundary friction affects the reproducibility of the relative positioning of the mirrors and the focusing adjustment. For this reason, the presence of a gap between the mirrors is necessary, and its effect on the focusing cannot be neglected. We have calculated the Fresnel integral in the cases of input Gaussian beams with different waists, namely w_0_ = 2.5 mm and 5 mm for the two wavelengths λ = 1 mm and 3 mm for three separations, namely 0 mm, 0.2 mm and 0.4 mm. The results are shown in the plots of Figure 5 and show that in all cases, the peak intensity decreases as the separation increases, the decrease being more pronounced at lower waists.

It is worth noting that even at a distance of only 4% of the beam diameter, i.e., 0.2 mm, and w_0_ = 5 mm, the intensity decreases by 12% approximately for both wavelengths. This finding suggests that for w_0_ = 2.5 mm, the positioning must be accurate to within a few tens of microns in order to keep the focusing efficiency close to maximum within a few percent variation. As the beam diameter increases to 10 mm, Figure 5 shows that the positioning tolerance is significantly relaxed as the peak intensity drop remains less than 5%. We conclude that for a Gaussian input with a beam waist w_0_ = 5 mm, an inter-mirror offset s of the order of ~200 μm is required to limit the decrease in the focusing factor to within 8%. Reducing the beam waist to w_0_ = 2.5 mm still allows well-defined focusing, albeit with a slightly lower focusing factor, but the inter-mirror offset should be kept within ~50 μm to limit the loss to a similar percentage, which is a not so tight condition to achieve with conventional mechanical movements.

## 3. Experimental Tests

### 3.1. Principle of the Measurement

Aiming at verifying the bimirror behavior, experimental simulations were performed with a setup including a visible laser source and a CCD camera, since THz instrumentation is not currently available in our laboratory. The validity of these simulations was theoretically ensured by the invariance property of the Fresnel integral just described in the previous paragraph. However, as described in Section 3.3, a preliminary verification of this procedure was possible by using a rented THz facility. We used a He-Ne laser (wavelength λ_vis_ = 0.633 μm) as a source and measured the calculated intensity focal distribution at THz frequencies for three different wedge angles. 

The optical setup is shown in Figure 6. The waist of the input probe beam (0.55 mm) is expanded to a beam waist w_0_ = 5 mm and sent to the bimirror probe. This consists of two custom rectangular mirrors with optical flatness λ/8 at = 633 nm. One of the two mirrors (M1) is mounted in a fixed position, while the other (M2) is mounted on a two-axis gimbal that can be moved in the x-y plane via a two-axis stage with 10 μm resolution. The mirrors are first brought into contact; then, M2 is moved laterally to create a small gap of about 0.1 mm to allow its free rotation, which is performed with micrometric precision. The gap can also be monitored by measuring the light power passing through it. The input beam is split by the bimirror into two beams which travel along an ABCD folding path to a screen.

The two beams interfere to produce a system of fringes whose visible number depends on the relative tilt between the mirrors, which is controlled by the gimbal-mounted mirror M2. The fringes are aligned vertically by using the tilt movements of M2 around the horizontal axis to minimize the angle between the mirror normals. The intensity distribution on the observation screen is recorded by a CCD camera and measured in real time by image processing software. The gimbaled mirror is moved angularly about the vertical direction until an absolute maximum of light intensity of the central fringe is obtained, i.e., the focus condition. In this situation, M1 and M2 form an angle 180−2α, where α is the wedge angle between the mirrors and the input plane as described in Figure 1 and the distance D is the magnified value of the focal length f_eq_ as reproduced with the visible radiation (see the following Section 3.2). 

The sequence of the described operations is shown in Figure 7. When the mirrors are coplanar, the light distribution reproduces the input Gaussian light intensity distribution with a slight distortion due to the presence of the gap. As the tilt of M2 is increased, the central fringe becomes more intense and the process is iterated until it reaches maximum brightness.

The angle α can be calculated in a first approximation by measuring the distance d between two adjacent maxima, as suggested by the theory of two-beam interferometers [18]. In fact, β is the angle between the beams and can be calculated to the first order as β = λ_vis_/d. Since β = 4α_vis_, the wedge angle required to produce the focal maximum on the screen can be easily calculated using the formula:(3)$$\alpha_{vis}=\frac{\lambda_{vis}}{4d}$$

Actually, this value of *α_vis_* has to be adjusted because of the diffractive effects. However, we have verified that this variation is of minor importance and increases with the wavelength to a maximum of 7.3% at *λ =* 3 mm.

Equation (3) gives the value α_vis_ which, according to the considerations in the Modeling section, namely, corresponds to the equivalent wedge angle α at a given THz wavelength *λ*, i.e.,:(4)$$\alpha =\alpha_{vis}\cdot M_{\lambda }$$

The focal distance *f_eq_* will be thus given by the following: (5)$$f_{eq}=D\cdot M_{\lambda }^{-1}$$

### 3.2. Experimental Results and Discussion

The transverse intensities of the focal distributions corresponding to three different distances D are shown in Figure 7. The distances between the central maximum and the maximum of the first lateral lobe correspond to wedges of α_vis_ = 57 μrads, 100 μrads, and 131 μrads, which are calculated using Equation (3) with the distances p, q, r shown in the figure. We arbitrarily simulated the focusing conditions by considering wedge angles decreasing with frequency, since this corresponds to the largest DOF for a given frequency (see the Modeling section). For example, for the wavelength λ = 3 mm (0.1 THz, M_λ_ = 4761), we consider a wedge angle α_vis_ = 57 μrads and D = 12 m and M_λ_ = 4761. 

Considering the transformation rule of Equation (4), these parameters correspond to the simulation of the focusing of the THz radiation with a wedge angle α = 0.27 rads. The corresponding theoretical transverse intensity distribution at the focal length f_eq_ = D/M_λ_ = 3.4 mm is shown in Figure 8d, (see Table 3). The calculated DOF is 6.6 mm, while the focal width w = 1.6 mm, which agrees well with the experimental value, can be extracted from Figure 8a. 

Similarly, well-defined peaked intensity distributions are found at distances of 7 m and 5.5 m. The traces in Figure 8b,c refer to the simulation of the focusing of the radiation at 0.33 THz (α = 0.16 rads) and 0.63 THz (α = 0.1 rads). The corresponding theoretical intensity plots calculated in the transverse plane at the axial focal positions are shown in Figure 8e,f. The increase in focal length of the bimirror compared to that obtained with a COE is more evident in Figure 9, which shows the 2D plots modeling the experimental situation at THz frequencies. The quantitative performance is summarized in Table 3. In all cases, the DOF expected for the bimirror is higher than that expected for the equivalent lens with an increase corresponding to ~14 times in the worst case of 0.1 THz.

Although the agreement between theory and experiment is always satisfactory, in all cases, it can be seen in Figure 8 that the sidelobes of the experimental transverse traces increase slightly with respect to the theoretical minima, even approaching zero at the nominal y-focus position f_eq_, which in the experimental situation corresponds to a distance D_nom_ = f_eq_∙M_λ_ from the instrument. The main reason for this slight discrepancy is the difficulty of placing the focal fringe at the fixed detection distance D. As can be seen in Table 3, the difference between the experimental best focus distance D and the theoretical one, Dmin, is only 12% of the DOF. As a consequence, it is not easy to locate the focus position with high precision, as would be expected for a long-focusing device where the intensity varies slowly with respect to the axial coordinate due to the robustness to diffraction. 

The evidence for this long focusing effect is also given by the difference of the focusing factor η calculated at the theoretical distance D_max_ and at the experimental best focus position D. This difference ∆ η is indeed very small, with a worst case change in ∆η = 2. 4% at 0.1 THz, while even smaller variations are calculated for 0.33 THz (∆η = 0.5%, η = 3.5) and 0.63 THz (∆η = 0.8%, η = 4.3), confirming the good potential of the proposed device as a focusing tool in the sub-THz range.

The long-focus property was also tested directly by simulating the propagation 0.1 THz radiation. The distance D was reduced from 12 to 5 m, i.e., a difference of 7 m, corresponding to a defocus of the 0.1 THz radiation of 7000/M_λ_ = 1.47 mm, obtaining the traces shown in the upper part of Figure 10. The intensity decrease is 15.6%, which is in good agreement with the expected theoretical value of 13.7%. In order to better evaluate the long focusing property of the bimirror, we compared its performance with that of an equivalent cylindrical optical element, which for greater convenience was a cylindrical lens. We measured the loss of maximum intensity when the observation screen is displaced axially by an amount of the same order as the expected depth of focus. We used a cylindrical lens with a focal length of 5 m (0.2 D), which focuses the incoming radiation into a spot with a measured FWHM of 0.81 mm.

The corresponding depth of focus is calculated using the conventional formula DOF = ±1.22·λ·(f#)^2^, where f# is the f-number of the lens, giving DOF = 490 mm. After moving the screen 0.5 m away from the device, the central maximum is reduced by about half, as shown in Figure 10 (lower part), while the FWHM increases to 1.1 mm. 

This result must be compared with the performance of the bimirror, for which a much larger displacement of 7 m is required to produce a 16% reduction, indicating its superior depth of focus compared with that produced by a spherical positive equivalent COE. 

The positive effect introduced by the bimirror can be further appreciated when it is combined with a focusing COE, which in this case is a lens with focal length f = 5.5 m (in Figure 5 inserted at a distance of about 15 cm in front of the bimirror), as shown in Figure 11a.

Such a long focal length was chosen to produce a focal spot of sufficient width to be reliably detected by the imaging system. In this case, the input field E(x_i_) undergoes a double phase change due to the lens and the bimirror, and it is modified in the analytical form shown in the red frame of Figure 11. The successive propagation is described by the integral of Equation (1) with the correct substitution of the phase terms. 

It is worth noting that the integral remains invariant with the correct rescaling of the focal length. In other words, the system simulates an optical layout consisting of a THz source and a positive lens whose focal length is f_THz_ = f·M_λ_^−1^, and the other distances and the wedge angle are scaled accordingly. For example, considering a source operating at 0.6 THz (λ = 0.5 μm, M_λ_ = 793), the corresponding focal length and defocus are transformed to 6.9 mm, while the wedge angle is increased from 2.2 × 10^−4^ to 0.18 rads. 

The effect of a defocus of −20 cm (−0.25 mm at 0.6 THz) is clearly visible in the photographs shown in Figure 11b,c. While the lens alone (Figure 11b) produces a clear decrease in spot intensity with defocus, the defocus effect is greatly reduced when the bimirror is present. The effect can be quantitatively deduced from the median curves shown in the plots of the same figure.

While in the first case the reduction is 25%, in the second case, it is reduced to 2.5%, indicating the significantly higher performance of the bimirror lens assembly. A similar advantage of the simultaneous use of a conventional positive component and a long-focus device is reported, for example, in [13]. 

Our results compare favorably with others reported in the literature. For example, the cylindrical super-oscillating lens (CSOL) theoretically studied for industrial surveillance and reported in [19], at 0.1 THz frequency, has a DOF higher than that reported in this work, but the FWHM of the focal line is higher, namely 2.52 mm, with a consequent lower maximum intensity. Moreover, the fabrication of the device requires the realization of an opaque grating of straight lines, which causes a transmission loss of about 25%. The parallel plate waveguide device in [13], operating in the 0.35–0.4 THz frequency range, was reported to have a DOF of about 20 mm and a focal width of 1.8 mm, while in our case, we have a DOF of about 9.5 mm in this band and a width of less than 1.0 mm with an advantage in terms of light intensity against a loss in depth of focus. The design of this device has been implemented to favor its tunability in other THz spectral windows, but this implies the manufacture of metal waveplates with different shapes and an increase in sensitivity to mechanical assembly with frequency, whereas in our case, the tunability is obtained simply by changing the wedge angle through the angular movement of a mirror. Moreover, while in the former case, the transmission depends on the ohmic losses, which in turn depend on the frequency, in our case, this dependence is significantly reduced, since the reflection of the mirrors is essentially constant over the whole THz range. The problem of long THz focusing has also been solved by assembling conventional systems with rotational symmetry [20]. In that work, a DOF of 60 mm is reported with a corresponding focal width of 2.8 mm at 0.1 THz with similar problems in terms of focusing performance and lack of tunability compared to our solution. This problem has also been addressed with more exotic geometries, such as metasurfaces [21], but their application is currently limited to solving specific problems requiring sub-mm wavelengths with fabrication issues that increase with frequency, and their performance (DOF = 8.4 mm, focal spot width w = 0.7 mm at 0.75 THz) is similar to our results. Again, no evidence of tunability is reported.

Some applications require a very large focal depth [11], in which case an interesting solution based on a cascade of lens-axicon doublets has been theoretically studied and successfully tested with a DOF close to 1 m and a focus width of 9.9 mm. However, the real applicability of this solution is questionable due to the high loss of the optical chain, which is about 2.7%. By reducing the number of doublets in order to increase transmission, the performance of such solutions is closer to that of our device.

### 3.3. Preliminary Comparative Tests with THz Radiation

In order to check the consistency of the measurements made with visible light, we performed some preliminary checks to validate this procedure.

The laser source was a TeraSmart THz Menlosystem spectrometer whose broadband emission, consisting of ps pulses, ranges from 0.1 to 5 THz. It was filtered around 1 THz (λ = 0.3 mm). The output beam was expanded to a nominal diameter of 2w = 4 mm and then sent to the two-mirror device (Figure 12a). 

The wedge angle was set to 0.04 rads and 0.03 rads, and the corresponding position of the real focus was calculated to be at distances from the bimirror wedge of 9.1 mm and 10.8 mm, respectively. The focus was imaged by a pair of TPX Tydex lenses (f = 25 mm, nominal focal length of the doublet f_d_ = 12.5 mm, diameter = 25.4 mm) in a 2f_d_ − 2f_d_ configuration with unit magnification, corresponding to a distance of the lenses from the bimirror of 25 mm and an equal image distance.

The choice of this configuration requires a few comments. Due to the diffractive effects, the conventional geometric–optical approximation for image formation, which assumes a δ-shaped impulse response, is no longer valid. Therefore, it can be difficult to obtain a good reproduction of the focal point. To this end, we calculated the focal length required to produce an image whose shape reasonably resembles that of the focal point. In fact, we found that choosing a focal length of 12.5 mm for the rely system, using a wavelength of 0.3 mm, produces only small deformations, as shown in the simulation in Figure 13. 

The equivalent configuration at λ = 633 nm was then constructed using a beam of the same diameter. The wedge angle was then scaled using the spectral factor M_λ_ = 476 (see Section 2) from 0.04 to 84 μrads and from 0.03 to 63 μrads, while the screen distance (see Figure 6) was set at 4 3 m and 5.1 m. The results shown in Figure 12 consist of the images obtained with the THz spectrometer using the two different values of the wedge angle (Figure 12e,g) and the images obtained with the setup of Figure 6 (Figure 12d,f). These intensity distributions can be compared with the theoretical normalized intensities calculated at 1 THz and shown in Figure 12b,e for α = 0.04 rads and α = 0.03 rads, respectively. Both the visible and THz images show a slightly narrower main lobe for the higher wedge of 0.04 rads, which is in agreement with the theoretical light distributions superimposed on the images in Figure 12d,f. This is accompanied by a slight narrowing of the main lobe and by a slight outward displacement of the secondary lobes. 

Although a more dedicated analysis should be carried out to further validate the congruence of the measurements, these results are a first validation of the proposed way to mimic the behavior of the bimirror at THz frequencies (but the underlying reasons could be extended to other devices) with more common sources whose radiation propagates at much higher frequencies. 

## 4. Conclusions

In this paper, we report a device capable of focusing a Gaussian beam with a frequency in the range of 0.1–1 THz over a large distance. The device consists of two planar mirrors that split the input beam into two superimposed beams by using the motion of one of the mirrors around two rotational axes. The relative angle between the mirrors and the width of the input Gaussian beam determine the depth of focus and the intensity maximum of the focus. We have studied its performance theoretically and experimentally for a given diameter of the input beam and have shown that thanks to this arrangement, the phase change necessary to form the optical beam into a long focus, even at lower THz frequencies, can be obtained with high values of the wedge angle. However, too large an angle reduces the depth of focus, so a trade-off between focusing efficiency and the need for a long depth of focus is generally required. Alternatively, a relay optical line is required to move the focus away from the device but still allow the sample to be placed far enough away from the device surface to be illuminated. 

The experimental tests were performed with a visible laser source by exploiting an invariance property of the 2D Fresnel diffraction integral. This property enables us to reproduce the propagation characteristics of the THz radiation in the propagation direction by stretching the axial coordinate, while in the transverse direction, the beam shape is not affected by the change in wavelength. However, this approach has been validated by preliminary tests with a THz laser source, which provided a first proof of the validity of the experimental simulation method. Theory and experiments show that our device achieves performances similar to those obtained with other techniques, but in our case, the device is very simple to build and can be easily tuned to the proper working needs by simply varying the angle between the mirrors. Moreover, since it works by exploiting the very high reflectivity in the low THz band, it avoids the typical transmission and dispersion problems of dielectric devices, which still limit the use of this radiation in several applications. 

Of all the possible future developments, we believe that reducing the size of the device is probably the most attractive. Indeed, we have shown that the use of Gaussian beams with waists of a few mm maintains a good focusing effect throughout the sub-THz band, so mirrors of millimeter dimensions are a possible goal. In this sense, the presented work represents a proof of concept that should be adequately addressed with a properly dedicated design.

## Figures and Tables

**Figure 1 micromachines-15-00715-f001:**
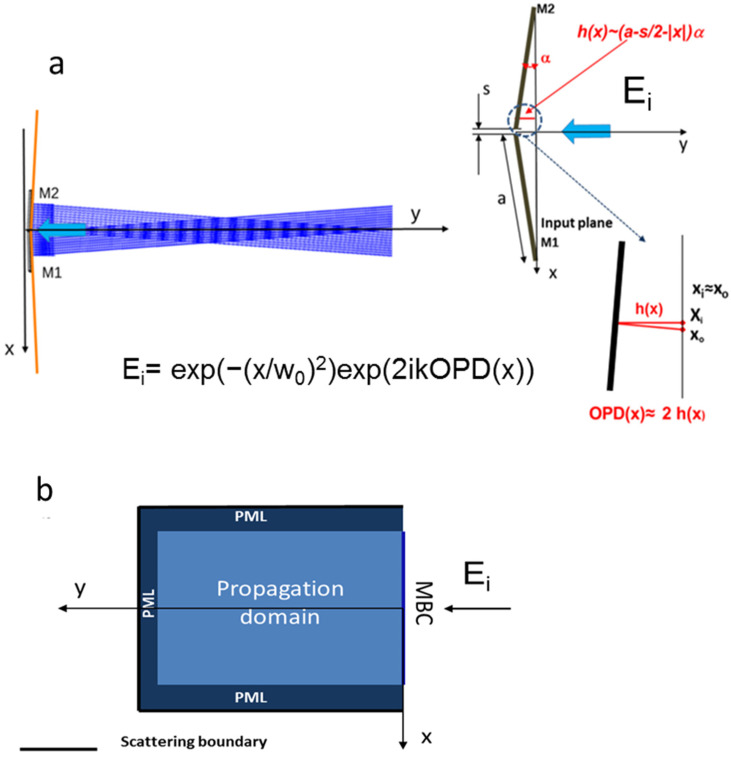
(**a**) (**left**) Ray tracing illustrating the working principle of the device and (**right**) its geometrical features. x_i_, x_o_ are the input and output abscissas, respectively, which are similar due to the thin device approximation. (**b**) Structure of the domain used to implement the COMSOL model. The input domain E_i_ is described analytically in Figure 1a.

**Figure 2 micromachines-15-00715-f002:**
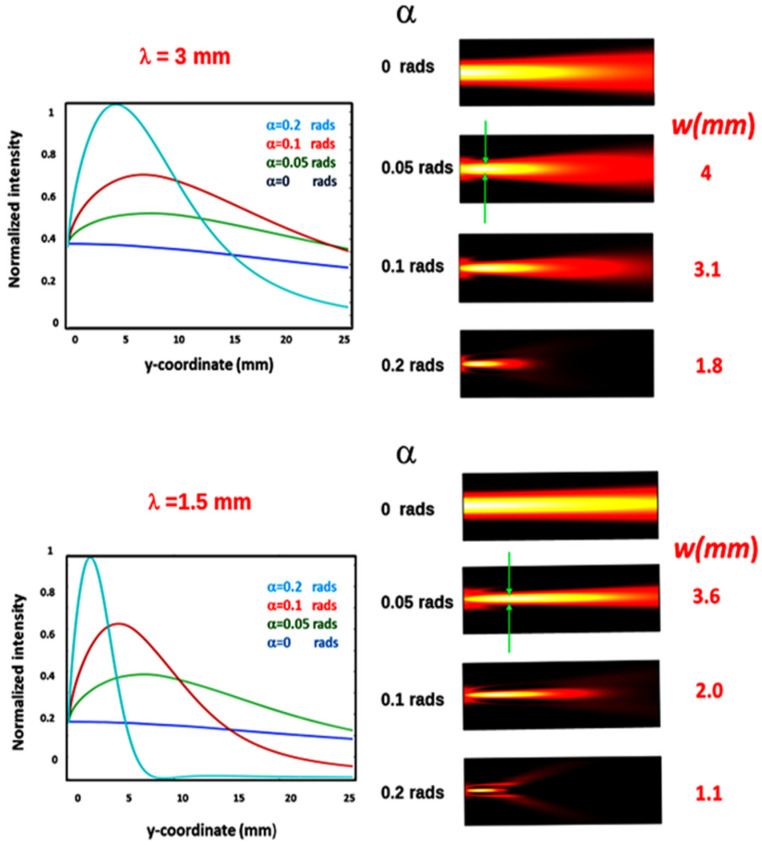
(**Upper part**, **left**): axial intensity distribution for a Gaussian input field at 0.1 THz frequency with waist w_0_ = 5 mm for different wedge angles. (**Right**): 2D light intensities indicating the higher light confinement for increasing wedge angles, provoking the increase in the focusing efficiency and the decrease in focal width reported on the right side. (**Bottom**): the same for 0.2 THz radiation. Note that at 0.2 rads, due to the lower diffraction effects, there is a clear reduction in the axial intensity and the upsetting of two separate lobes. For the lower frequency of 0.1 THz, the effect is strongly smeared at the same wedge angle.

**Figure 3 micromachines-15-00715-f003:**
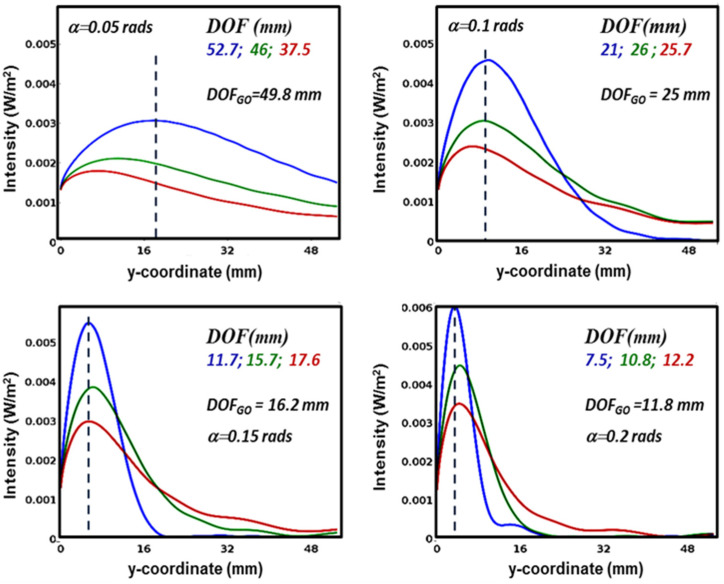
Axial intensities of the radiation at three different frequencies (0.1 THz, 0.2 THz, 0.3 THz) for the four values of the wedge angle α indicated in each figure, also showing the depths of focus as defined in the text for each frequency, marked with the appropriate color, as well as the depth of focus predicted by geometrical-optical considerations (DOF_GO_).

**Figure 4 micromachines-15-00715-f004:**
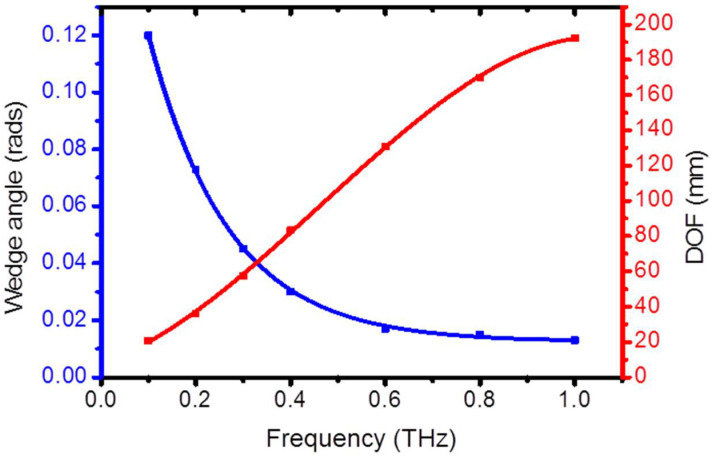
Wedge angle *α* (blue) and DOF (red) for a bimirror that focuses the light with efficiency *η* = 2. The lines represent interpolations of the calculated data (points). The input field beam waist is w_0_ = 5 mm.

**Figure 5 micromachines-15-00715-f005:**
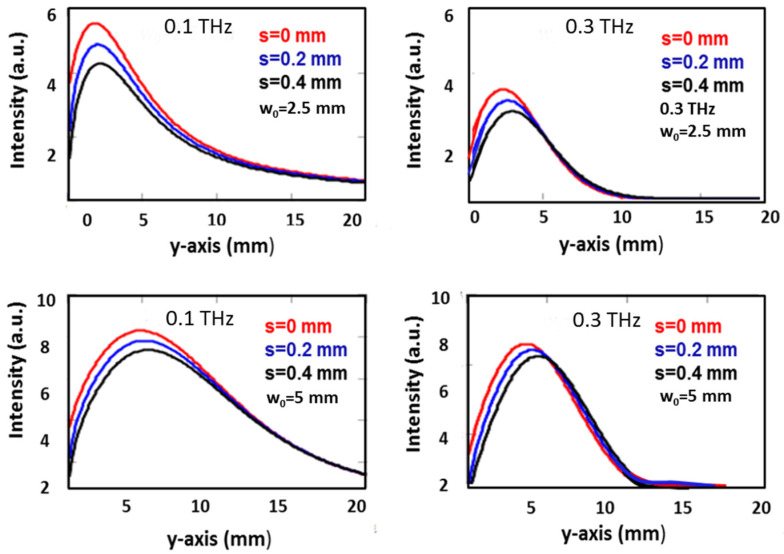
Effect of the mirrors separation *s* on the axial intensity. For each figure of the panel, both beam waist and frequency are shown.

**Figure 6 micromachines-15-00715-f006:**
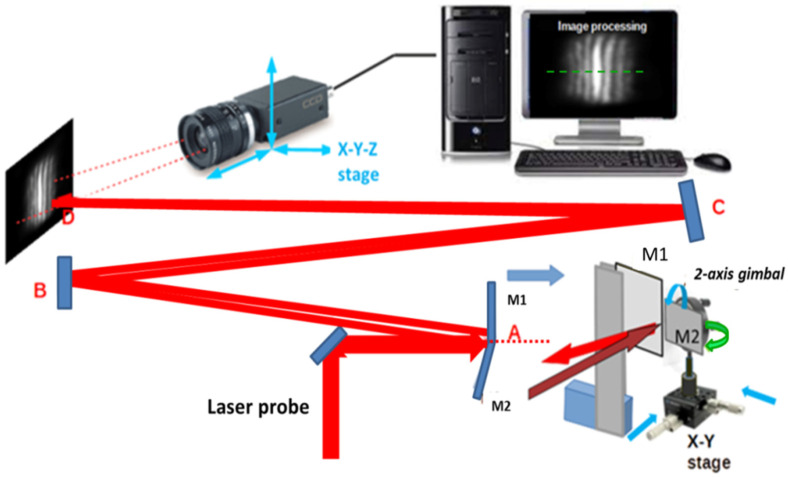
Experimental layout. An expanded (9.1×) laser beam is sent to the bimirror, which splits it into two beams that interfere on the screen along a path ABCD whose length is D. The mirror M1 is in a fixed position, while the mirror M2 is placed at a distance s from M1 by means of an x-y stage and rotated to superimpose the two beams. A small fraction of the input radiation passes through the aperture between the mirrors (dashed red line), allowing further control of the separation s. The intensity distribution is recorded by a CCD camera and processed by image processing software, allowing the light intensity to be recorded in real time on a given horizontal line of pixels (dashed green line).

**Figure 7 micromachines-15-00715-f007:**
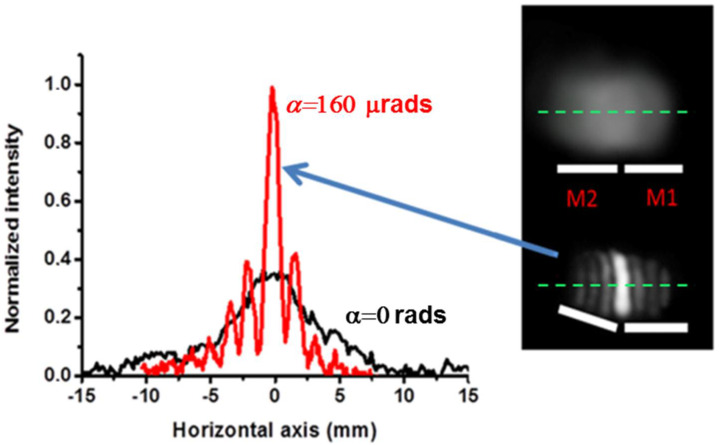
Focusing of the input beam at a distance D from the bimirror. The photograph on the right shows the light distributions before (top) and after (bottom) rotation of the movable mirror M2 through a calculated angle of 1.6 × 10^−4^ rads. The experimental traces on the center line of the light distributions are recorded and superimposed on the left of the figure. There is a clear focusing effect, indicated by the red trace, with an increase in light intensity in the center.

**Figure 8 micromachines-15-00715-f008:**
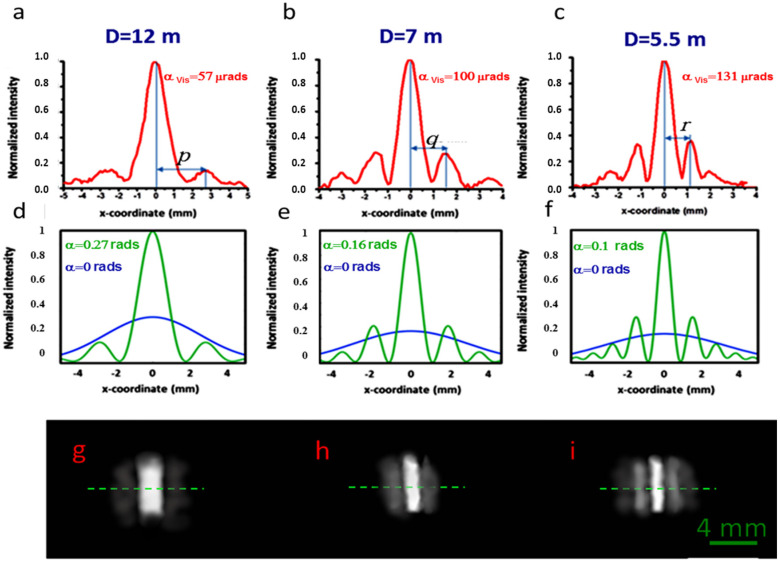
Light intensity distributions at different distances D from the bimirror. (**a**–**c**): Experimental traces recorded on the median line (green dashed lines in **g**–**i**). The plot (**a**) reproduces the intensity of radiation at 0.1 THz frequency with f_eq_ = 3.4 mm, (**b**) 0.3 THz and f_eq_ = 5 mm, (**c**) 0.63 THz and f_eq_ = 6.8 mm. The angles α_vis_ are calculated using Equation (4) in the text. The theoretical green plots (**d**–**f**) are the transversal intensity distributions calculated using the above angles α in the nominal axial position of the foci (see Table 3), while the blue curves represent the intensity distribution of the input beam. (**g**–**i**): Images of the light distributions at the experimental focal positions of (**g**) 12 m, (**h**) 7 m and (**i**) 5.5 m. See the text for more details on the calculation of the wedge angles α.

**Figure 9 micromachines-15-00715-f009:**
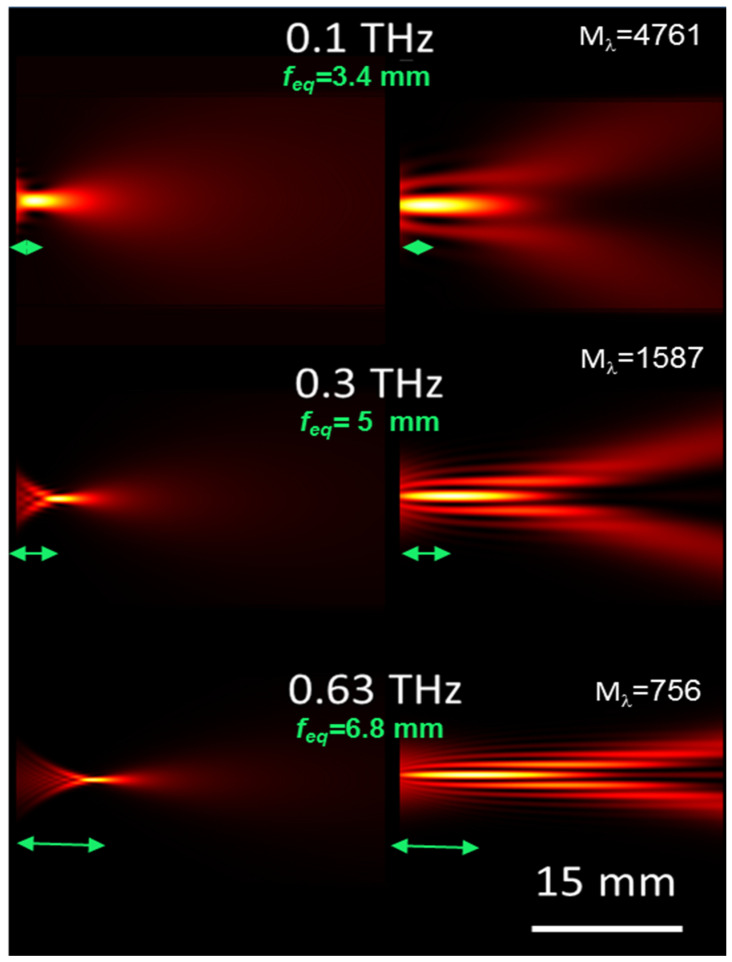
Two-dimensional (2D) plots representing the modeling of the focusing experiments whose results are reported in Figure 8. **Left column**: plots of the focused field obtained with COE. **Right column**: focal distributions obtained with the bimirrors with the same equivalent focal lengths (indicated in the figure with green arrows).

**Figure 10 micromachines-15-00715-f010:**
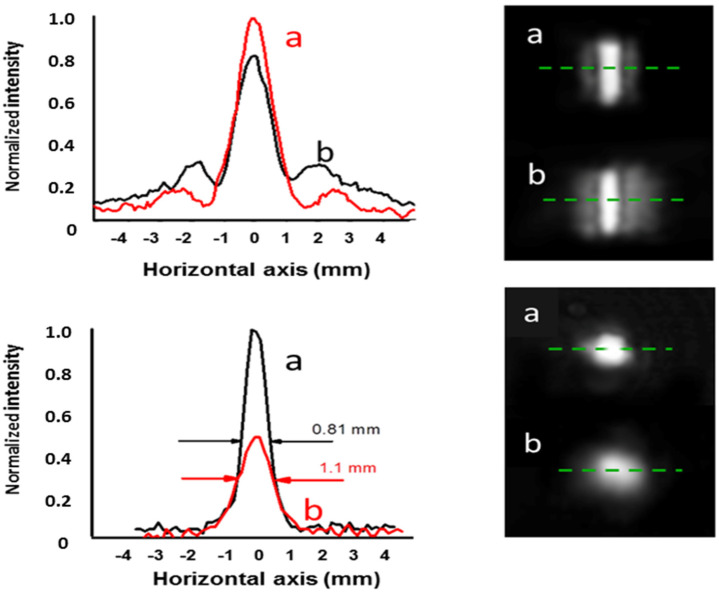
Comparison of the focusing of the bimirror (**top**) and an equivalent COE with the same focal length (**bottom**). While the top plot and the corresponding images on the right clearly show a substantial persistence of the maximum intensity for a displacement of 7 m, the focus of the lens is more susceptible to diffraction, as the maximum intensity drops by 50% for a displacement of only 0.5 m.

**Figure 11 micromachines-15-00715-f011:**
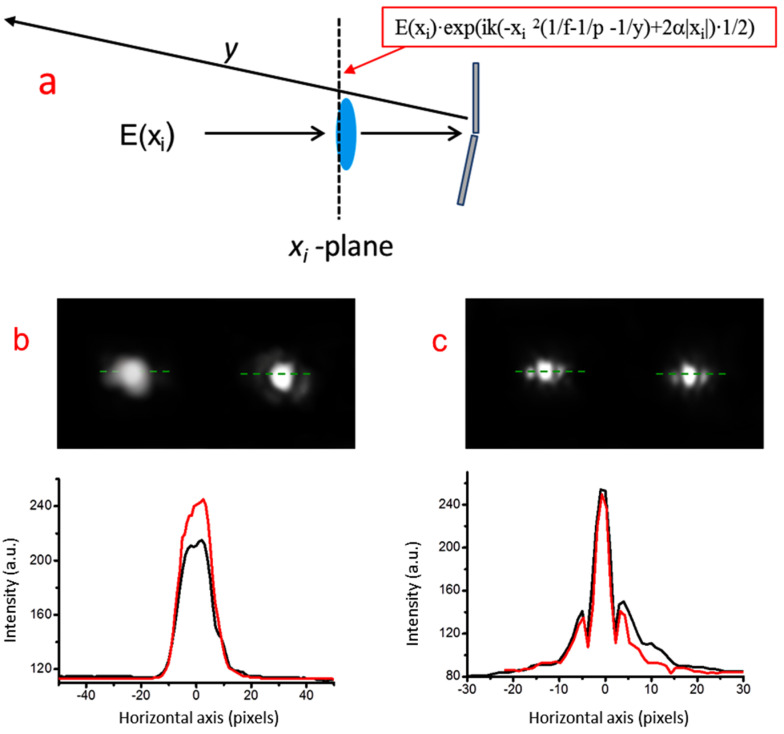
(**a**) Focusing of the probe radiation onto a target at a distance z from the bimirror due to the combined effect of a positive lens (focal length = 5.5 m) and the bimirror. The field distribution input E(x_i_) is subject to the phase shift introduced by the lens plus bimirror (estimated wedge angle α = 2.2 × 10^−4^ rads) and takes the analytical form shown in the red frame. It then propagates to the object plane at distance y. In the lower part, photos (**b**,**c**) show the focused probe generated by the lens alone and by the lens plus bimirror, respectively, at nominal focus (red lines) and at −20 cm defocus (black lines).

**Figure 12 micromachines-15-00715-f012:**
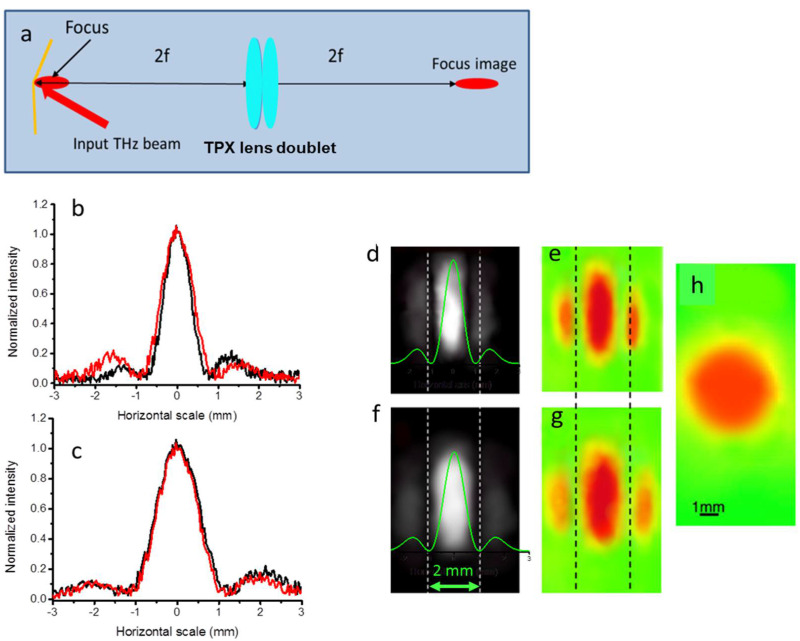
(**a**) Schematic of the THz imaging system. (**b**) Normalized transversal intensity measured for the wedge angle α = 0.04 rads. Red trace: transversal scan on the THz image (**e**), black trace: scan on the visible image (**d**). (**c**) Normalized transversal intensities measured for the wedge angle α = 0.03 rads. Red trace: scan on THz image (**g**), black trace: scan on visible image (**f**). The vertical dashed lines are a guide for the eye to better see the shape variation of the pattern and the displacement of the secondary lobes. (**d**,**f**). Images taken with the setup of Figure 6 at distances of 4.3 m and 5.1 m for the scaled α = 84 μrads and α = 63 μrads, respectively. (**h**) The image of the input THz beam. The green traces superimposed on the images (**c**,**f**) are the theoretical intensities calculated at 0.5 THz.

**Figure 13 micromachines-15-00715-f013:**
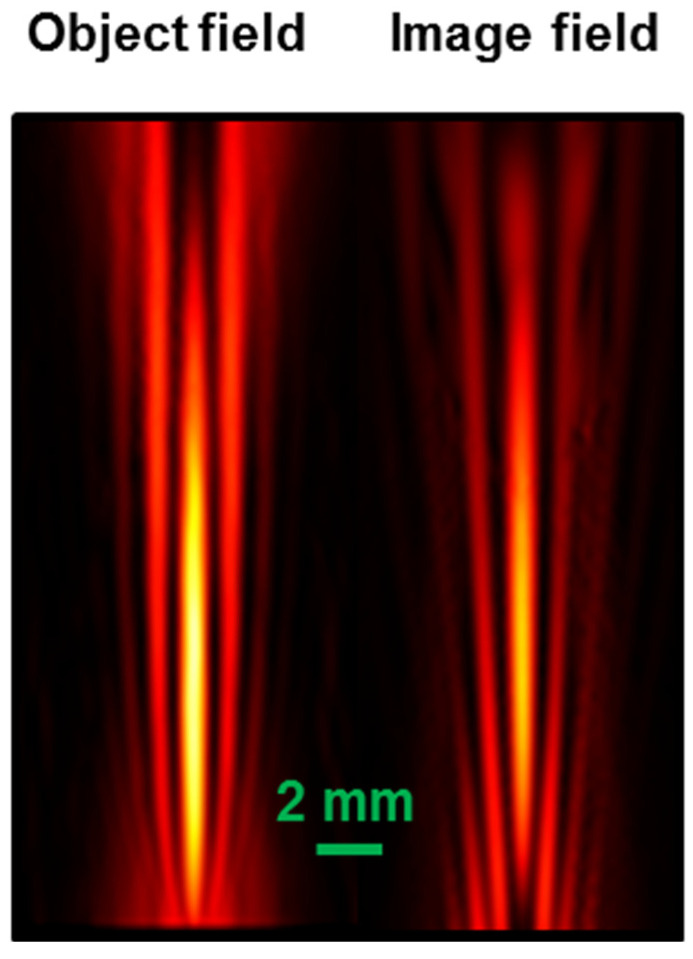
Object field (left) and its simulated image (right), consisting of a focused Gaussian beam with waist *w*_0_ = 2 mm and frequency 1 THz, considering a wedge angle α = 0.04 rads, produced with a positive optic with focal length 12.5 mm placed at 25 mm from the bimirror wedge in a configuration with unit magnification. The distortion in the upper part of the image field is due to the definition of the mesh of the COMSOL code, which was the best trade-off between resolution and computational time. The difference of intensity is due to the reflections on the lens surfaces.

**Table 1 micromachines-15-00715-t001:** Focusing performances of a bimirror with wedge angle *α* = 0.2 rads and of a COE with the same focal length *f_eq_*. Input beam waist *w*_0_ = 5 mm.

	f_eq_ (mm)	DOF (mm)	w (mm)	D_G_	DOF_G_	η (%)
0.1 THz	4.8	12.2	1.8	1.92	1.9	2.3
0.33 THz	4.0	10.8	0.72	0.53	0.45	4
0.66 THz	2.4	7.5	0.32	0.16	0.082	4.8

**Table 2 micromachines-15-00715-t002:** Focusing performances of a bimirror with wedge angle α = 0.1 rads and of a COE with the same focal length *f_eq_*. Input beam waist *w*_0_ = 5 mm.

	f_eq_ (mm)	DOF (mm)	w (mm)	D_G_	DOF_G_	η (%)
0.1 THz	6.9	25.7	2.9	2.76	4	1.6
0.33 THz	9.7	26	1.34	1.28	2.6	2.8
0.66 THz	7.3	21	0.71	0.49	0.74	4

**Table 3 micromachines-15-00715-t003:** Parameters describing the focusing of the bimirror in the experimental simulation of the behavior of radiations at three THz frequencies. The reported DOFs are calculated considering the values of α = α_vis_.

	α_vis_ (mrads)	*w* (mm)	D (m)	M_λ_	f_eq_ (mm)	DOF (mm)	DOF_G_ (mm)	D_nom_ (m)	Δη (%)
0.1 THz	57	1.4	12	4761	3.4	6.6	0.46	15.7	2.4
0.33 THz	101	1.2	7	1587	5	8.5	0.61	7.9	0.5
0.63 THz	131	0.81	5.5	762	6.8	11.5	0.53	5.2	0.8

## Data Availability

Data will be delivered upon request.

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
