# Peer review of "Tunable Device for Long Focusing in the Sub-THz Frequency Range Based on Fresnel Mirrors"

_micromachines, 2024, doi:10.3390/mi15060715_

Round 1
Reviewer 1 Report (Previous Reviewer 1)
Comments and Suggestions for Authors
The manuscript can be published in Micromachines
Comments on the Quality of English LanguageThe manuscript can be published in Micromachines
Author Response
Dear reviewer,
thank you for the revision. We are glad to see that you found the paper good as presented and doesn’t need revisions.
Best regards,
- Giancarlo Margheri
Reviewer 2 Report (New Reviewer)
Comments and Suggestions for Authors
Document attached

Moderate editing of English language required
Author Response
Dear referee,
thank you for the accurate revision work.
Herebelow please find the answers to your comments. Apart the corrections added in the figures, those ones in the text have been highlighted in pale blue.
1.In the abstract there is a line with two dots, and the term "bi-mirror" is improperly written. I would also rewrite the portion about the setup used for the experimental results, omitting justifications for not using THz equipment but providing the motivation for using a different setup.
We fixed throughout the whole paper the term “bi-mirror” with “bimirror” present in the literature. We changed the final statement in the Abstract the motivation for justifying the use of instrumentation working in the visible.
2.The introduction is well-written, with updated and well-explained references. The introduction to the work is articulated clearly. I would rewrite "a priori" without using italics.
We rewrite the term “a priori” as indicated
3 Figure 1 and its explanation are quite confusing. The angle alpha is presented differently in Figure 1a and Figure 1b, and the written part doesn't aid comprehension, especially since values of -2alpha and 2alpha are then given. Also, the presentation of the Gaussian electric field needs improvement, as it is referred to in the text as Figure 1b, while it's actually depicted in Figure 1c. It would be beneficial to provide a formal definition of the output abscissa, not just a numerical one.
We restructured Fig.1 by dividing it into two parts. We defined formally in the caption the input-output abscissas.
- Before mentioning the formulas used in the theory, it's advisable to include references.
We included Ref.16 to justify the use of the Fresnel integral for cylindrical coordinates: Sheppard C.J.R., Cylindrical lenses- focusing and imaging: a review. App.Opt., 2013, 52,4,538-545.
5.I suggest separating Figure 1c from the rest and referring to it in the text when discussing the Comsol simulations. Provide a more detailed explanation of why this model is being used and how it simulates what is depicted in Figures 1a and 1b.
As anticipated, we separated Fig.1 from the rest. We included a more accurate description of the the simulation model, why it has been used and how it has been adapted to the solution of the propagation model of the device shown in Fig.1a
6.Figure 3 could be improved by including the values of alpha considered for the different figures.
We included the values of a in the figures of the panel.
- In Figure 5, adding both frequency and beam waist values directly onto the graph would allow for immediate visualization of the differences. Meanwhile, the legend for separation values could be omitted for all four figures to avoid redundancy.
We modified the panel of Fig.5 adding frequency and beam waist in each figure.
- If I understand correctly, from Figure 2 to Figure 5, all results are from simulations. If this is correct, why are they included within the theory paragraph? It would be beneficial to add a "Simulation Results" section for clarity and better organization
We introduced the section 2.2 “Simulation results”.
- Figure 7 requires a legend for the two different curves (red and black).
We included the legends close to the plots.
10.In Figure 8, why do the theoretical green curves have intensity values greater than 1? It would also be beneficial to add a legend for the different curves in this figure.
We modified the curves as required.
- Line 396 should read "Figure 9," not "Table 9."
Actually, , we couldn’t find any correspondence between line 396 and “Table 9” using the lines numbering. Thus, we checked with automatic research for the terms “Table”and “9”, but once more we didn’t succeeded in finding the two terms associated. Rather, we found a “Fig.9” correctly located.
- The font size and the distribution of values in Table 3 should be improved and standardized to match what was done in Tables 1 and 2.
Tables 1,2,3 have been matched in the same font format Calibri 18. Any mismatch in the font size is due to the presence of a higher number of columns in Tab.3, that has obligated to reduce the dimensions of the letters.
- Given the experimental results presented on the THz system, the initial justification for the cost of this system in that section becomes irrelevant. It would be more appropriate to rewrite the justification for why a visible laser system was used instead.
Other than in the Abstract, in the beginning of the “Experimental test” section, we described the reason why we used a simulation method with a visible source.
- The caption of Figure 13 should be rewritten as "Object field (left) and its simulated image (right),...."
We complied accordingly.
- The conclusions are well-written
Thank you.
Round 2
Reviewer 2 Report (New Reviewer)
Comments and Suggestions for Authors
The paper is now ready for the publication. All the raised questions have been replied. Good luck with your work with THz spectrometer
This manuscript is a resubmission of an earlier submission. The following is a list of the peer review reports and author responses from that submission.
Round 1
Reviewer 1 Report
Comments and Suggestions for Authors
The manuscript of Giancarlo Margheri and Tommaso Del Rosso is devoted to the focusing of sub-THz radiation by the Fresnel mirrors. I believe that the novelty and impact of the manuscript does not fulfill the criteria of Micromachines for the following reasons:
1. Authors derived the scaling factor for the central wavelength of radiation to transmit from the sub-THz range to the visible one. The experiment was performed using the He-Ne laser (not with the sub-THz source). Such an experiment has a level of a practical task for college students. The generalization of results obtained to the sub-THz optics is problematic. The first obvious reason for this is the excellent quality of He-Ne lasers providing Gaussian beam with the plane phase. In contrast, the THz sources like quantum-cascade laser emit the radiation with ring-like intensity distribution and complicated phase in space [Phys. Rev. Lett. 96, 173904 (2006)].
2. The Comsol simulations are quite simple and does not account for the specific features of the sub-THz fields as well.
3. The paper with very close results was published by Authors recently [Micromachines 14, 1939 (2023)].
Thus, I recommend Editor to reject the manuscript.
Reviewer 2 Report
Comments and Suggestions for Authors
This is an interesting work, but the feasibility of the method in the terahertz band is not well demonstrated experimentally, so the readers may have some doubts about the validity of the method. It is suggested to select a suitable terahertz source and detector for testing, and give the experimental results of terahertz band.